# Postoperative analgesic effects of paravertebral block versus erector spinae plane block for thoracic and breast surgery: A meta-analysis

**Chang Xiong** [1], **Chengpeng Han**[2], **Dong Zhao**[1], **Wenyong Peng**[1], **Duojia Xu**[1], **Zhijian Lan**[1] *

1 Department of Anesthesiology, Affiliated Jinhua Hospital, Zhejiang University School of Medicine, Jinhua, Zhejiang, China, 2 Department of Rehabilitation, Jinhua Maternal and Child Health Care Hospital, Jinhua, Zhejiang, China

☯ These authors contributed equally to this work.
* jylanzhijian@163.com

**Data Availability Statement:** All relevant data are within the manuscript and its Supporting Information files.

## Abstract

### Background

Paravertebral block (PVB) is the most recognized regional anesthesia technique after thoracic epidural anesthesia for postoperative analgesia in thoracic and breast surgery. Erector spinae plane block (ESPB) is a recently discovered blocking technique, and it has evidenced excellent postoperative analgesia for breast and thoracic surgery with fewer adverse reactions. However, there are controversies about the postoperative analgesic effects of the two analgesic techniques.

### Objective

To assess the analgesic effects of PVB versus ESPB in postoperative thoracic and breast surgery.

### Methods

We systematically searched PubMed, Cochrane Library, EMBASE, Web of Science, and ScienceDirect databases up to April 5, 2021. The primary outcome was postoperative pain scores. Secondary outcomes included: opioid consumption, additional analgesia, postoperative nausea and vomiting (PONV) 24 hours post-operation, and the time required for completing block procedure. This study was registered in PROSPERO, number CRD42021246160.

### Results

After screening relevant, full-text articles, ten randomized controlled trials (RCTs) that met the inclusion criteria were retrieved for this meta-analysis. Six studies involved thoracic surgery patients, and four included breast surgery patients. Thoracic surgery studies included

**Funding:** This study was supported by the Science and Technology Research Project of Jinhua (http://kjj.jinhua.gov.cn/art/2021/4/2/art_1229317787_3832377.html), grant number (2021-4-004). Xiong Chang was responsible for this. The sponsors decided to publish this study.

**Competing interests:** The authors have declared that no competing interests exist.

all of the outcomes involved in this meta-analysis while breast surgery did not report pain scores at movement and additional analgesia in 24 hours post-operation. For thoracic surgery, PVB resulted in significant reduction in the following pain scores: 0–1 hours (MD = -0.79, 95% CI: -1.54 to -0.03, P = 0.04), 4–6 hours (MD = -0.31, 95% CI: -0.57 to -0.05, P = 0.02), and 24 hours (MD = -0.42, 95% CI: -0.81 to -0.02, P = 0.04) at rest; significant reduction in pain scores at 4–6 hours (MD = -0.47, 95% CI: -0.93 to -0.01, P = 0.04), 8–12 hours (MD = -1.09, 95% CI: -2.13 to -0.04, P = 0.04), and 24 hours (MD = -0.31, 95% CI: -0.57 to -0.06, P = 0.01) at movement. Moreover, the opioid consumption at 24 hours post-operation (MD = -2.74, 95% CI: -5.41 to -0.07, P = 0.04) and the incidence of additional analgesia in 24 hours of the postoperative course (RR: 0.53, 95% CI: 0.29 to 0.97, P = 0.04) were significantly lower in the PVB group than in the ESPB group for thoracic surgery. However, no significant differences were found in pain scores at rest at various time points postoperatively, and opioid consumption at 24 hours post-operation for breast surgery. The time required for completing block procedure was longer in the PVB group than in the ESPB group for thoracic and breast surgery, and the incidence of PONV between the two groups showed no significant difference.

## Conclusion

The postoperative analgesic effects of PVB versus ESPB are distinguished by the surgical site. For thoracic surgery, the postoperative analgesic effect of PVB is better than that of ESPB. For breast surgery, the postoperative analgesic effects of PVB and ESPB are similar.

## Introduction

Acute postoperative pain is a common complication of most surgeries, especially for thoracic and breast surgeries. And ineffective alleviation of postoperative pain may produce a series of detrimental acute and chronic complications such as: anxiety, hemodynamic disturbances, immunity imbalances, increased myocardial oxygen consumption, myocardial injury, etc [1–3]. For thoracic surgery, it may also cause lung infections, hypoxemia, and prolonged hospital stay [4,5]. Therefore, finding a safe, effective and convenient analgesic drug or method for pain management after thoracic and breast surgery has become the focus of many anesthesiologists. In the past, thoracic epidural anesthesia (TEA) was regarded as the gold standard for postoperative analgesia after thoracic and breast surgery due to its superior analgesic effect [6]; however, complications such as the high incidence of hypotension, respiratory depression, and potential spinal cord injury associated with TEA limit its wide range of applications [7]. In recent years, thoracic peripheral nerve block has gradually received attention and promotion with the cooperation of ultrasound technology, mainly including paravertebral block (PVB), erector spinae plane block (ESPB) and intercostal nerve block. Among them, PVB has been reported to have similarly effective analgesic effects as TEA, resulting in relatively fewer complications [8,9]. But PVB requires high technical proficiency. Otherwise, it will easily cause pneumothorax and neurovascular damage [10]. ESPB is actually a kind of interfascial block, greatly avoiding the damage of peripheral nerves and blood vessels. When local anesthetics injected into the deep surface of the erector spinae muscle and the surface of the transverse process, it can diffuse anteriorly into the adjacent paravertebral and inter-costal spaces, thus blocking the dorsal and ventral rami of the spinal nerves. So, the ESPB technique, along with

its significantly reduced block procedure difficulty, has earned clinicians' and researchers' interest, and studies have demonstrated the benefits of ESPB for reducing the demand for postoperative analgesic drugs and postoperative analgesic scores [11]. Meanwhile, a growing number of randomized controlled trials (RCTs) compared PVB with ESPB in terms of the postoperative analgesic effects and complications, but the results are not consistent. Motivated by the controversy, this meta-analysis aims to assess the postoperative analgesic effects of PVB and ESPB for thoracic and breast surgery.

## Materials and methods

The Preferred Reporting Items for Systematic Reviews and Meta-Analyses (PRISMA) recommendations [12] were followed in the preparation of this meta-analysis. The protocol of this study was registered prospectively at International Prospective Register of Systematic Reviews (PROSPERO) (number CRD42021246160). The detailed information of the protocol was shown in **S1 File**.

### Search strategy

Two independent researchers (Chang Xiong and Chengpeng Han) conducted the literature search process in PubMed, Cochrane Library, EMBASE, Web of Science, and ScienceDirect databases from the inception of each database to April 5, 2021. The search terms included the following: "paravertebral plane block", "paravertebral block", "PVB", "erector spinae plane block", "ESP block", "thoracic surgery", "thoracoscopic surgery", "thoracotomy", "modified radical mastectomy", "mastectomy" and "breast surgery". We made appropriate adjustments when searching different databases. The retrieval strategy in PubMed database is shown in **S2 File**.

### Selection criteria/Eligibility

The inclusion criteria were as follows: (1) RCTs; (2) Adult patients (over 18 years old) who underwent thoracic or breast surgery; (3) Interventional use of PVB for postoperative analgesia (PVB group); (4) Use of ESPB for postoperative analgesia (ESPB group) in the control group. The exclusion criteria were as follows: (1) Incomplete studies; (2) Unreported relevant outcomes.

### Data extraction

Relevant data, including author, year of publication, country, number of patients in each group, location of nerve block, local anesthetic dose, surgical approach, duration of surgery and outcomes were extracted independently from eligible articles by two researchers (Chang Xiong and Chengpeng Han). Attempts were made to retrieve raw data for continuous variables from the eligible articles if variables in the full texts were presented as median and range; however, if data could not be extracted, then, the median and range were transformed to the mean ± standard deviation (SD) [13,14]. WebPlotDigitizer was used to extract numerical data if data values were given in a graphical format [15]. Any disagreements arising from the entire process were arbitrated by a third experienced researcher (Zhijian Lan). The primary outcome was postoperative pain scores. The secondary outcomes included opioid consumption, additional analgesia, postoperative nausea and vomiting (PONV) at 24 hours post-operation, and the time required for completing block procedure.

### Quality assessment

The quality of included RCTs was assessed by two independent researchers (Chang Xiong and Chengpeng Han) using the Cochrane Collaboration Risk of Bias tool and the

Jadad Score [16,17]. Any disagreement was also arbitrated by a third researcher (Zhijian Lan). The articles were evaluated under Cochrane Collaboration Risk of Bias tool and Jadad Score. Cochrane Collaboration Risk of Bias tool includes six bias-based parameters: selection (random sequence generation and allocation concealment), performance (blinding of participants and personnel), detection (blinding of outcome assessment), attrition (incomplete outcome data), and reporting (selective reporting) biases, and each parameter was classified as "low", "unclear", or "high". Jadad Score (total 5 points) was defined according to three criteria: randomization (0–2 points), blinding (0–2 points), and statements of possible withdrawals (0–1 point). The study was considered low-quality if its Jadad Score was less than two and high-quality if its Jadad Score was greater than or equal to three.

## Statistical analysis

Continuous data on outcomes, including postoperative pain scores, opioid consumption at 24 hours post-operation and the time required for completing block procedure were presented as mean difference (MD) at 95% confidence interval (CI). The dichotomous data on outcomes, such as incidence of additional analgesia in 24 postoperative hours and PONV were expressed as the relative risk (RR) at 95% CI. The $\chi^2$ test and $I^2$ statistic were employed to estimate statistical heterogeneity across studies. The $I^2$ statistic was stratified into three levels: low-level (0–49%), moderate-level (50%–74%), and high-level (>75%). The fixed-effects model was used in the event of low-level heterogeneity; otherwise, a random-effect model was applied. For moderate-level and high-level of heterogeneity ($I^2$>50%), sensitivity analysis or subgroup analysis was performed. Sensitivity analysis was performed by omitting one study by turns. Subgroup analysis based on a priori hypothesis that is the analgesic effects of PVB and ESPB are related to surgical site. The Egger's test, as well as visual examination of the funnel plot were used to assess potential publication bias. RevMan (version 5.3; Cochrane Library, Oxford, UK) was used to perform meta-analyses, and STATA 14/MP (StataCorp., College Station, TX, USA) was used for conducting Egger's test (metabias module).

## Results

### Characteristics of the included studies

A total of 503 studies were identified during the initial search. Among which, 86 duplicates were removed. Through thorough review of the titles, abstracts, and full texts of the research studies, 10 RCTs [18–27] enrolled a total of 726 patients, were eventually selected for the final analysis. The flowchart of the study selection is shown in **Fig 1.** The selected studies were published between 2019 and 2020, and they were carried out in different countries, such as China, Turkey, Egypt, USA, and Japan. Six studies [18,19,21,25–27] involved thoracic surgery, including five thoracoscopic surgery and one thoracotomy. The other four [20,22–24] were conducted on adult females receiving breast surgery. Nine studies [18–22,24–27] assessed pain scores at different time points after surgery. According to these time points, we divided four time intervals to aggregate the pain scores data, which were 0–1, 4–6, 8–12 and 24 hours after surgery. For two pain score assessments at the same time interval, we extracted the first assessment data for final analysis. Thoracic surgery included all of the outcomes involved in this meta-analysis, while breast surgery did not reported pain scores at movement and the additional analgesia in 24 hours post-operation. All studies performed nerve blocks under ultrasound guidance. The general characteristics of all the RCTs are summarized in **Table 1**.

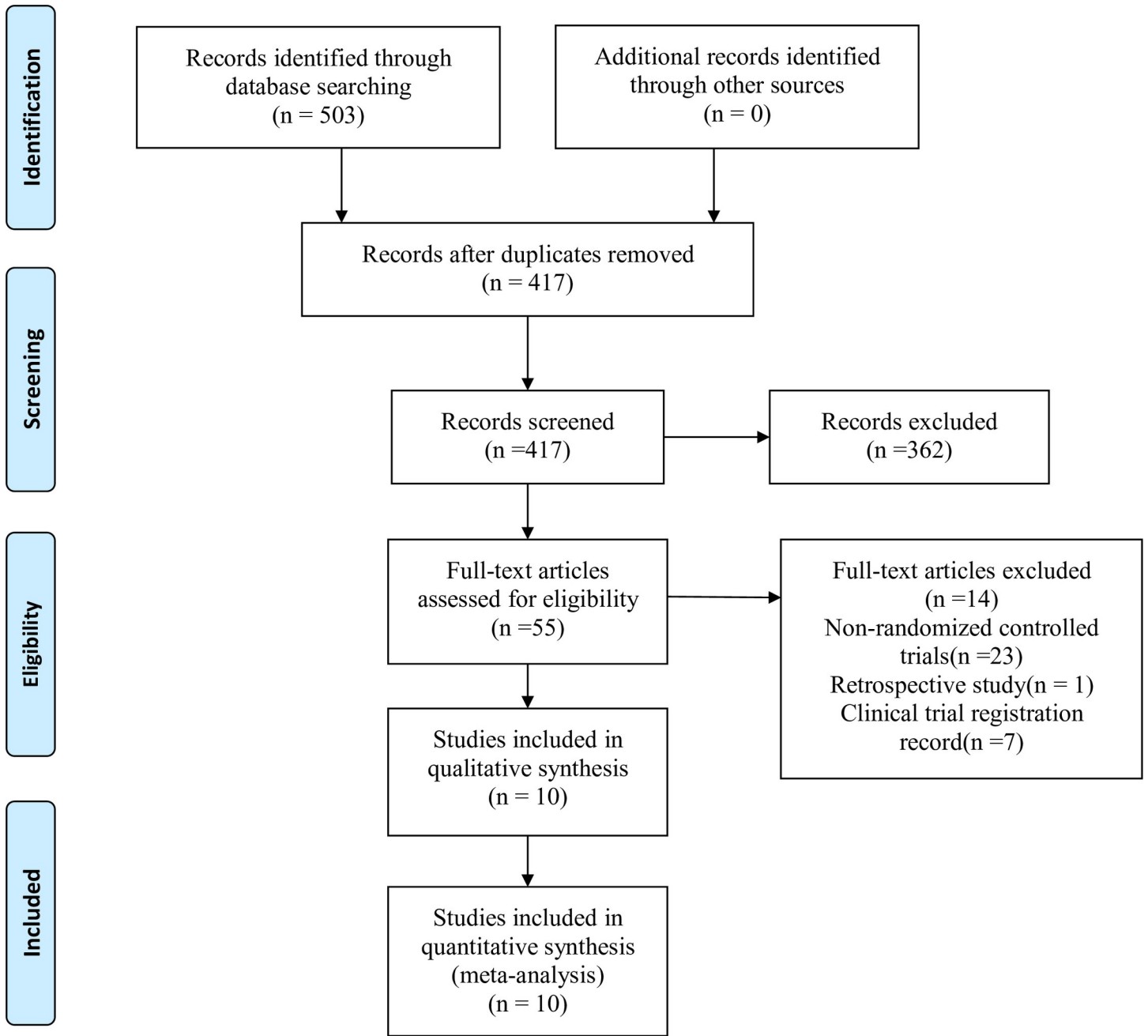

**Fig 1. PRISMA flow diagram summarizing retrieved, included, and excluded RCTs.** PRISMA indicates Preferred Reporting Items for Systematic Reviews and Meta-Analyses.

## Results of quality assessment

Six studies [18,20–24] displayed a high risk of bias arising from detection, attrition and reporting. The remaining four articles [19,25–27] had unclear risks of bias. The outcomes of risk assessment are presented in **Fig 2**. All trials were identified as high quality, according to the Jadad Score. The Jadad Score is shown in **Table 1**.

**Table 1. Characteristics of 10 included studies.**

| Study | Country | Jadad Score | Number (PVB/ ESPB) | Age(PVB/ ESPB) | Location | Local anesthetic dose | Surgery | Duration of surgery (min) | Time to assess pain scores | Outcomes |
|---|---|---|---|---|---|---|---|---|---|---|
| Chen N 2020 [18] | China | 5 | 24/24 | 51.6±10.4/ 53.3±11.6 | PVB at T5-T7; ESPB at T5 | 20 ml of 0.375% ropivacaine | VATS | 128.4±58.2/ 134.5±43.1 | At 0, 2, 4, 8, 24, 48 hours postoperatively | Pain scores; Opioid consumption; Additional analgesia; PONV |
| Çiftçi B 2020 [19] | Turkey | 4 | 30/30 | 47.53±10.43/ 47.33±10.21 | PVB at T5; ESPB at T5 | 20 mL of 0.25% bupivacaine | VATS | 125.86 ±17.67/ 135.50±29.13 | At 1, 2, 4, 8, 16, 24, 48 hours postoperatively | Pain scores; Opioid consumption; Additional analgesia; Block procedure time; PONV |
| El Ghamry MR 2019 [20] | Egypt | 3 | 35/35 | 41±11.8/37.7 ±12.9 | PVB at T5; ESPB at T5 | 20 ml of 0.25% bupivacaine | Breast surgery | 173.2±8.7/ 170±8.2 | At 0, 2, 4, 6, 8, 12, 18, 24 hours postoperatively | Pain scores; Opioid consumption; PONV |
| Fang B 2019 [21] | China | 5 | 46/45 | 59.39±9.95/ 61.73±9.32 | PVB at T5; ESPB at T5 | 20 mL of 0.25% bupivacaine | Thoracotomy lung surgery | 72.61±24.47/ 78.33±29.62 | At 1, 6, 24, 48 hours postoperatively | Pain scores; Opioid consumption; Block procedure time; PONV |
| Gürkan Y 2020 [22] | Turkey | 4 | 25/25 | 49.4 ± 7.25/ 49.08 ± 10.56 | PVB at T4; ESPB at T4 | 20 ml 0.25% bupivacaine | Breast surgery | 82.2 ± 22.54/ 89.4 ± 22.83 | At 1, 6, 12, 24 hours postoperatively | Pain scores; Opioid consumption; PONV |
| Moustafa MA 2020 [23] | Egypt | 3 | 45/45 | / | PVB at T4; ESPB at T4 | 20 ml 0.25% bupivacaine | Breast surgery | / | / | Opioid consumption; Block procedure time |
| Swisher MW 2020 [24] | USA | 3 | 50/50 | 55.4±17.6/ 55.4±11.7 | PVB atT2/ 4 or T3/5; ESPB at T3 or T4 | 25 ml 0.5% ropivacaine unilateral | Breast surgery | 75.7±47.3/ 77.9±31.3 | At 0, 24 hours postoperatively | Pain scores; Opioid consumption; Block procedure time |
| Taketa Y 2019 [25] | Japan | 5 | 40/41 | 67±8/70±7 | PVB at T4 or T5; ESPB at T4 or T5 | 20mL 0.2% levobupivacaine | VATS | 178.6 ± 28.2/ 179.3 ± 48 | At 1, 2, 4, 8, 12, 24, 48 hours postoperatively | Pain scores; Opioid consumption; Additional analgesia; PONV |
| Turhan Ö 2020 [26] | Turkey | 3 | 35/35 | 53.97±7.34/ 53.31±9.03 | PVB at T5; ESPB at T5 | 20mL of 0.5% bupivacaine | VATS | 101.71 ±24.55/97.71 ±43.05 | At 0, 1, 4, 12, 24, 36, 48 hours postoperatively | Pain scores; Opioid consumption; Additional analgesia; PONV |
| Zhao H 2020 [27] | China | 3 | 33/33 | 57 ± 6/59 ± 5 | PVB at T4 and T6; ESPB at T4 and T6 | 30 ml 0.4% ropivacaine | VATS | 107±30/121 ±58 | At 24, 48 hours postoperatively | Pain scores; Opioid consumption |

VATS: Video-assisted thoracoscopic surgery; PONV: Postoperative nausea and vomiting.

## Primary outcome: Pain scores at rest and movement at 0–1, 4–6, 8–12, and 24 hours after surgery

Six thoracic surgeries and three breast surgeries reported postoperative pain scores over time for PVB versus ESPB, respectively. At rest, the synthetic analysis performed by using a random-effect model indicated that compared to ESPB, PVB resulted in statistically significant reduction of postoperative pain scores at 0–1 hours (MD = -0.63, 95% CI: -1.12 to -0.14, P = 0.01, $I^2$ = 81%), 4–6 hours (MD = -0.23, 95% CI: -0.41 to -0.06, P = 0.009, $I^2$ = 16%), and 24 hours (MD = -0.33, 95% CI: -0.65 to -0.01, P = 0.04, $I^2$ = 78%) in the postoperative course (Fig 3). Those pain scores at 8–12 hours exhibited no significant difference between PVB and

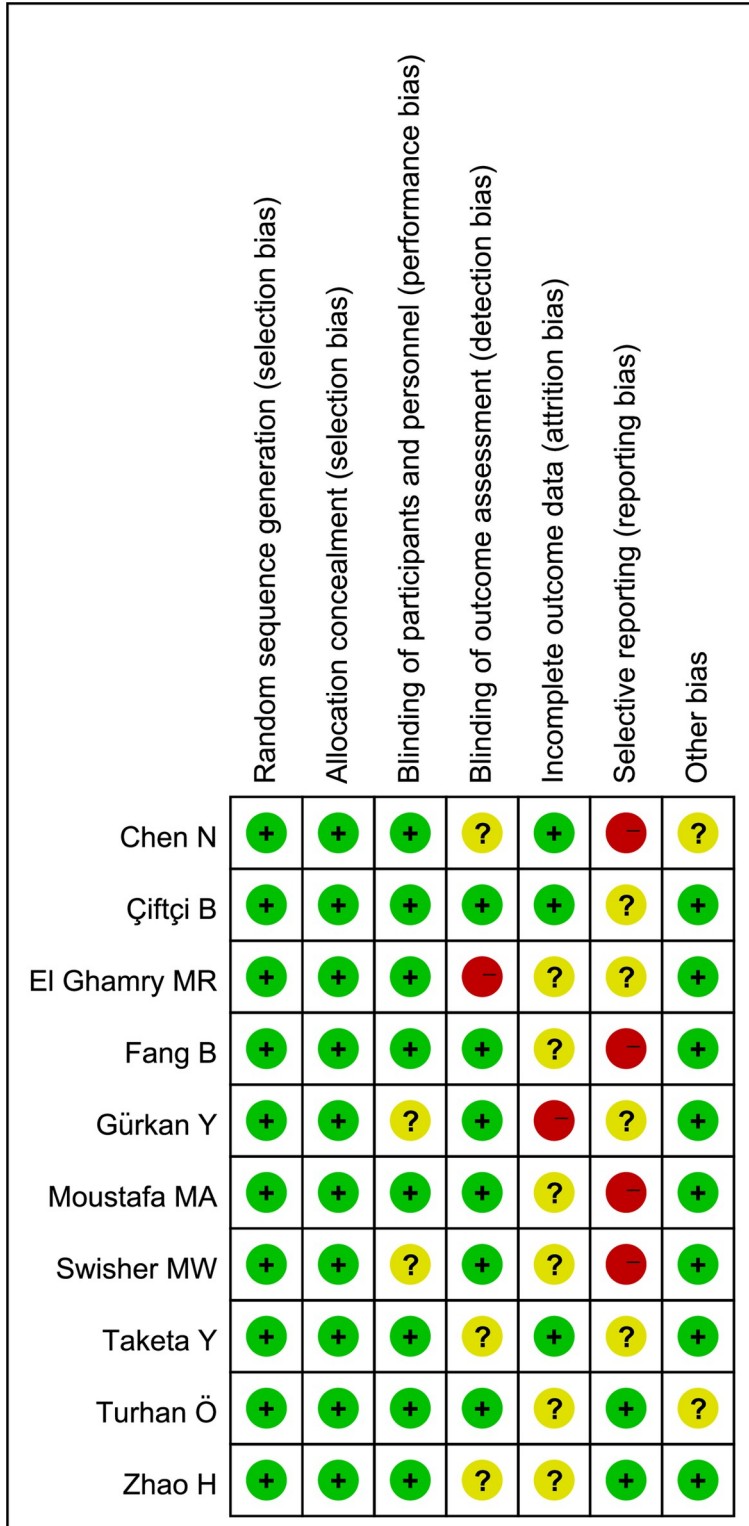

**Fig 2. Risk of bias assessment using Cochrane criteria.**

ESPB groups (MD = -0.45, 95% CI: -1.19 to 0.30, P = 0.24, $I^2$ = 89%; **Fig 3**). At movement, the pooled analysis revealed that when compared to ESPB, PVB resulted in significant statistical

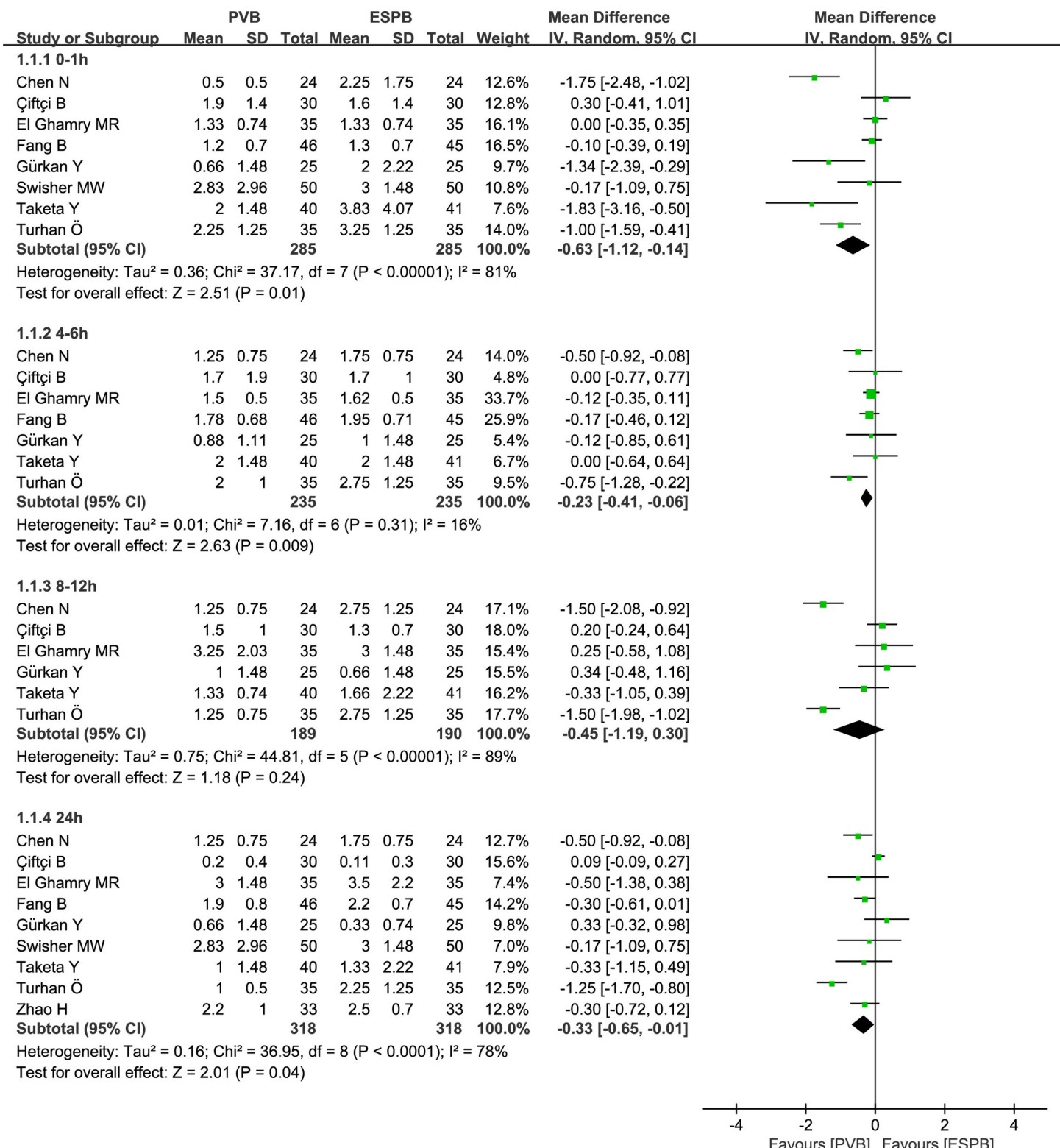

**Fig 3. Forest plot of postoperative scores for resting pain at 0–1, 4–6, 8–12, and 24 hours.**

reduction in pain scores at 4–6 hours (MD = -0.47, 95% CI: -0.93 to -0.01, P = 0.04, $I^2$ = 66%), 8–12 hours (MD = -1.09, 95% CI: -2.13 to -0.04, P = 0.04, $I^2$ = 91%), and 24 hours (MD = -0.31, 95% CI: -0.57 to -0.06, P = 0.01, $I^2$ = 49%). (Fig 4). No significant differences were observed at 0–1 hours between PVB and ESPB groups (MD = -0.65, 95% CI: -1.51 to 0.22, P = 0.14, $I^2$ = 88%; Fig 4). Considering the existence of moderate to high-levels of heterogeneity ($I^2$>50%), we conducted the subgroup analysis to assess whether the pain scores at rest were influenced by the surgical site (Fig 5). The results showed that compared to ESPB, PVB resulted in lower postoperative pain scores at 0–1 hours (MD = -0.79, 95% CI: -1.54 to -0.03,

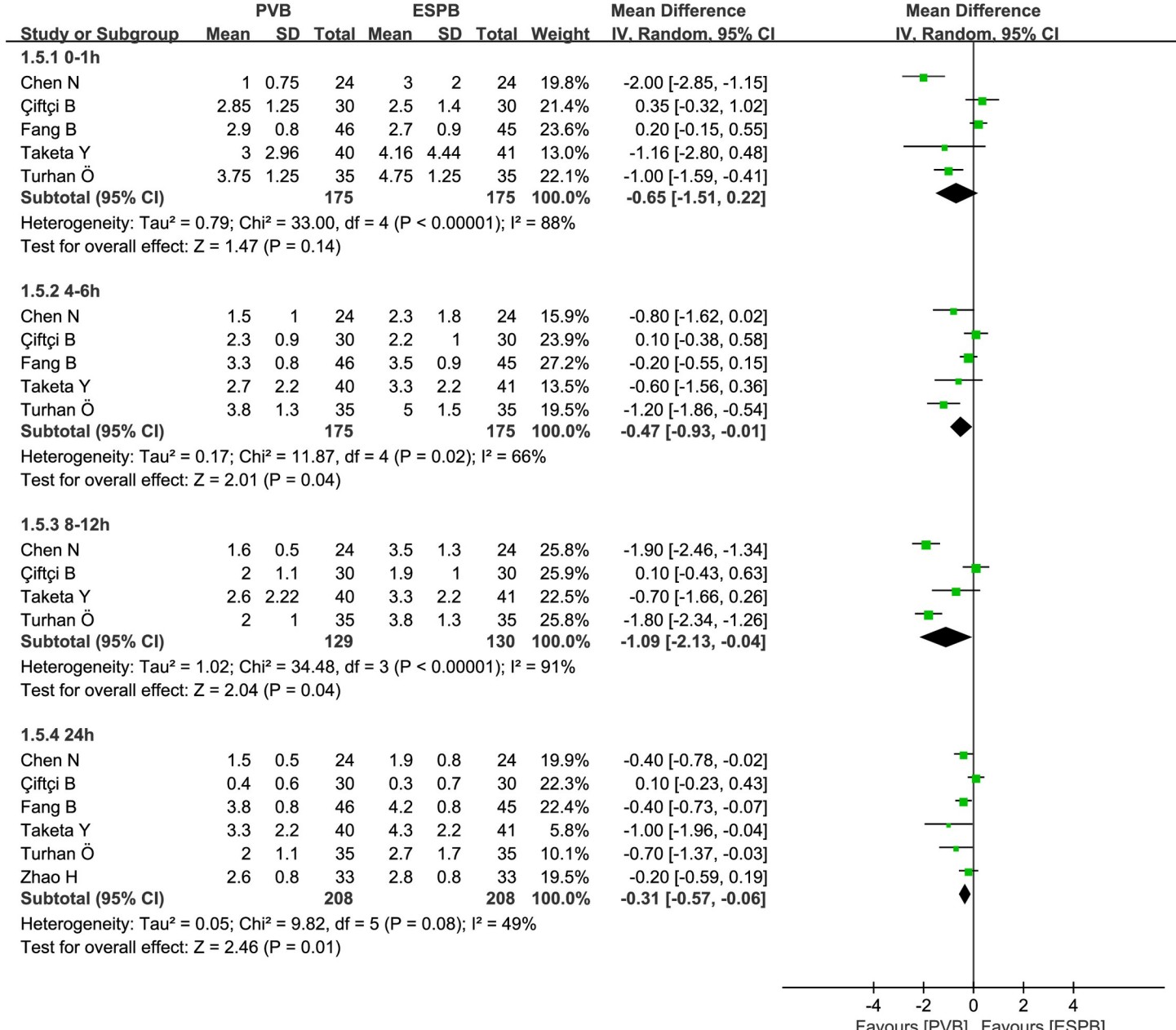

**Fig 4. Forest plot of postoperative scores for moving pain at 0–1, 4–6, 8–12, and 24 hours.**

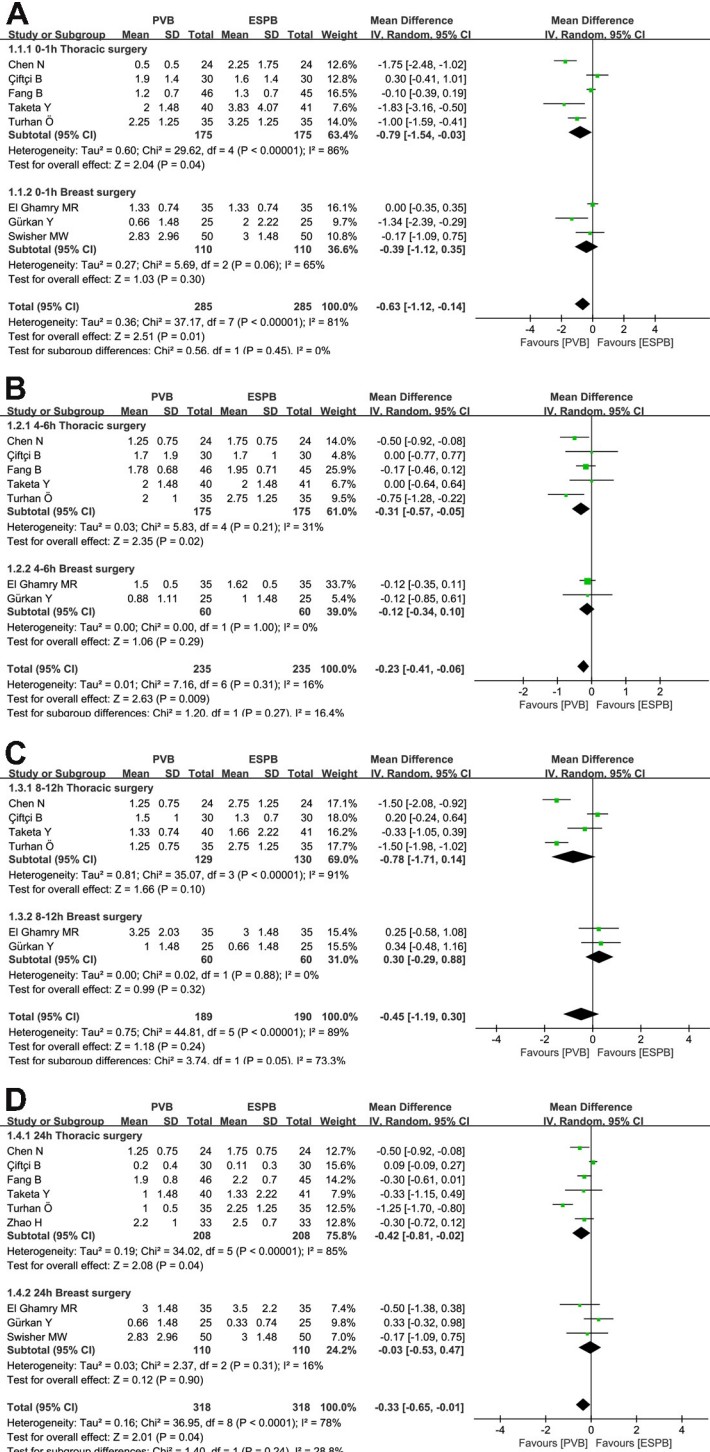

**Fig 5.** Forest plot of the subgroup analysis of postoperative scores for resting pain at 0–1 hours (A), 4–6 hours (B), 8–12 hours (C) and 48 hours (D).

P = 0.04, $I^2$ = 86%), 4–6 hours (MD = -0.31, 95% CI: -0.57 to -0.05, P = 0.02, $I^2$ = 31%), and 24 hours hours (MD = -0.42, 95% CI: -0.81 to -0.02, P = 0.04, $I^2$ = 85%) post-operation for

thoracic surgery, while no significant differences were reported at 8–12 hours (MD = -0.78, 95% CI: -1.71 to 0.14, P = 0.10, $I^2$ = 91%). No significant differences were observed at 0–1 hours (MD = -0.39, 95% CI: -1.12 to 0.35, P = 0.30, $I^2$ = 65%), 4–6 hours (MD = -0.12, 95% CI: -0.34 to 0.10, P = 0.29, $I^2$ = 0), 8–12 hours (MD = 0.30, 95% CI: -0.29 to 0.88, P = 0.32, $I^2$ = 0), and 24 hours (MD = -0.03, 95% CI: -0.53 to 0.47, P = 0.90, $I^2$ = 16%) between PVB and ESPB groups in breast surgery.

## Opioid (morphine equivalent) consumption at 24 hours after surgery

Five thoracic surgeries and four breast surgeries reported opioid consumption at 24 hours after surgery for PVB compared to ESPB, respectively. Random-effects model did not reveal any significant differences between PVB and ESPB groups (MD = -1.08, 95% CI: -2.67 to 0.51, P = 0.18, $I^2$ = 86%; **Fig 6A**). Subgroup analysis indicated that more opioids were required in the ESPB group than in the PVB group for thoracic surgery (MD = -2.74, 95% CI: -5.41 to -0.07, P = 0.04, $I^2$ = 86%), while no statistical difference existed between PVB and ESPB groups after breast surgery (MD = 0.31, 95% CI: -0.34 to 0.97, P = 0.35, $I^2$ = 1%) (**Fig 6B**).

## Incidence of additional analgesia in 24 hours post-operation

Four thoracic surgeries recorded additional analgesia in 24 hours post-operation for PVB versus ESPB. Pooled results revealed a higher incidence of additional analgesia in the ESPB group compared with that in the PVB group (RR: 0.53, 95% CI: 0.29 to 0.97, P = 0.04, $I^2$ = 62%; **Fig 7**). After performing the sensitivity analysis by omitting the study by Çiftçi B (2020) [19], no heterogeneity was found among the remaining trials ($I^2$ = 0), and the pooled results were unaltered.

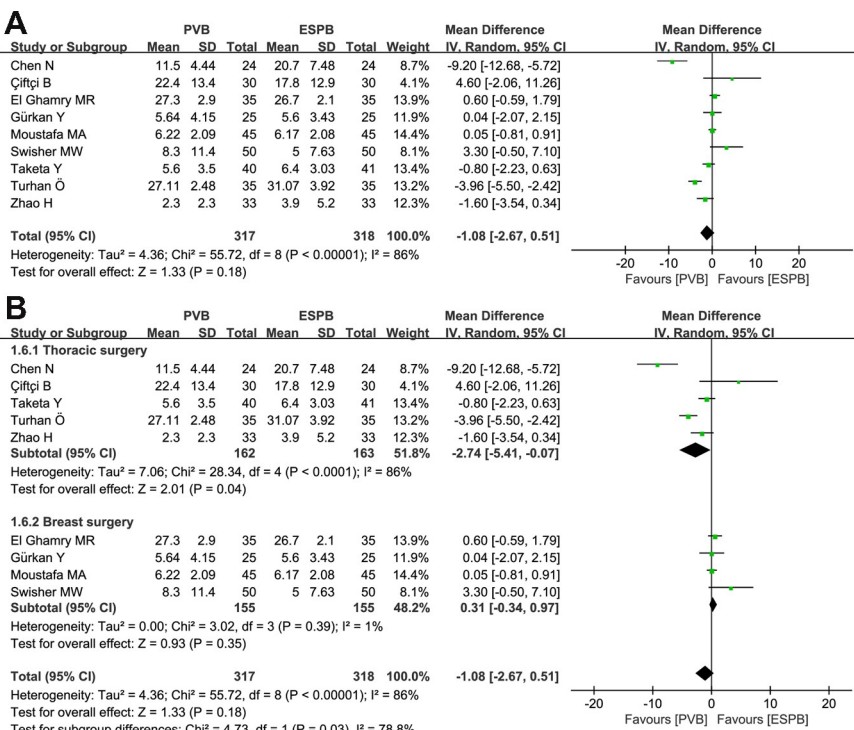

**Fig 6. Forest plots of the opioid consumption at 24 postoperative hours.** (A) Analysis of all data in the associated studies; (B) Subgroup analysis by differentiating thoracic surgery or breast surgery.

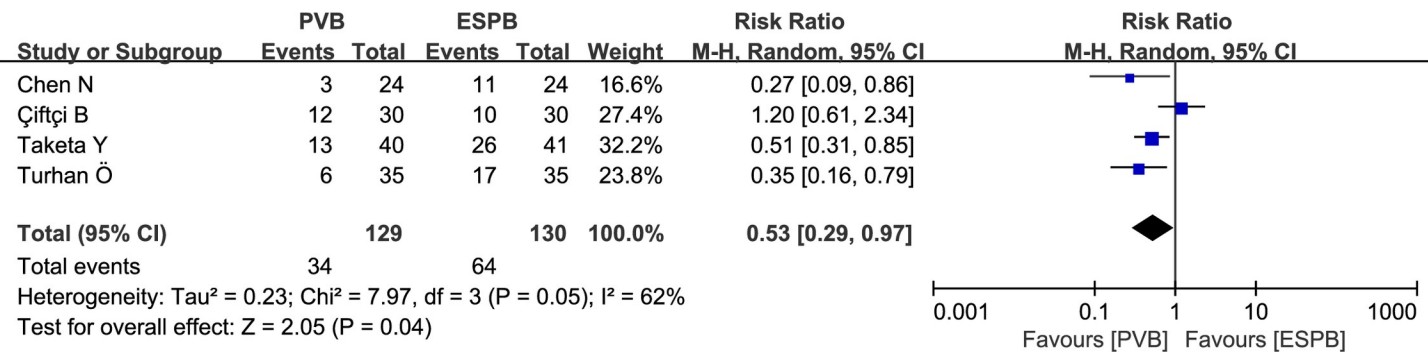

**Fig 7. Forest plots of the incidence of additional analgesia in 24 postoperative hours.**

## Time required for completing block procedure

Two thoracic surgeries and two breast surgeries analyzed the time (minutes) required for completing the block procedure process for PVB and ESPB, respectively. Random-effects model demonstrated that the time to complete the ESPB under the guidance of ultrasound was shorter than that in PVB (MD = 4.05, 95% CI: 2.95 to 5.14, P<0.00001, $I^2$ = 87%; **Fig 8A**). Subgroup analysis indicated that more time was required for completing the block procedure in the PVB group than in the ESPB group both for thoracic and breast surgeries (MD = 4.86, 95% CI: 2.90 to 6.82, P<0.00001, $I^2$ = 89%; MD = 3.29, 95% CI: 2.31 to 4.26, P<0.00001, $I^2$ = 71%; **Fig 8B**).

## Incidence of postoperative nausea and vomiting (PONV)

Five thoracic surgeries and two breast surgeries analyzed incidence of PONV for PVB and ESPB, respectively. The pooled analysis revealed that there was no significant difference between ESPB and PVB groups, with a low level of heterogeneity(RR: 0.81, 95% CI: 0.57 to 1.16, P = 0.25, $I^2$ = 20%; **Fig 9**).

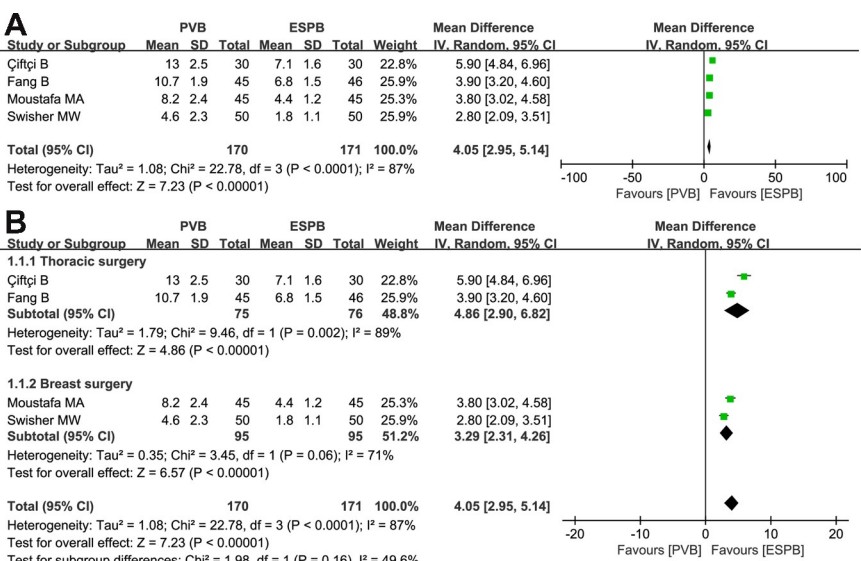

**Fig 8. Forest plots of the time required for completing block procedure.** (A) Analysis of all data in the associated studies; (B) Subgroup analysis by differentiating thoracic surgery or breast surgery.

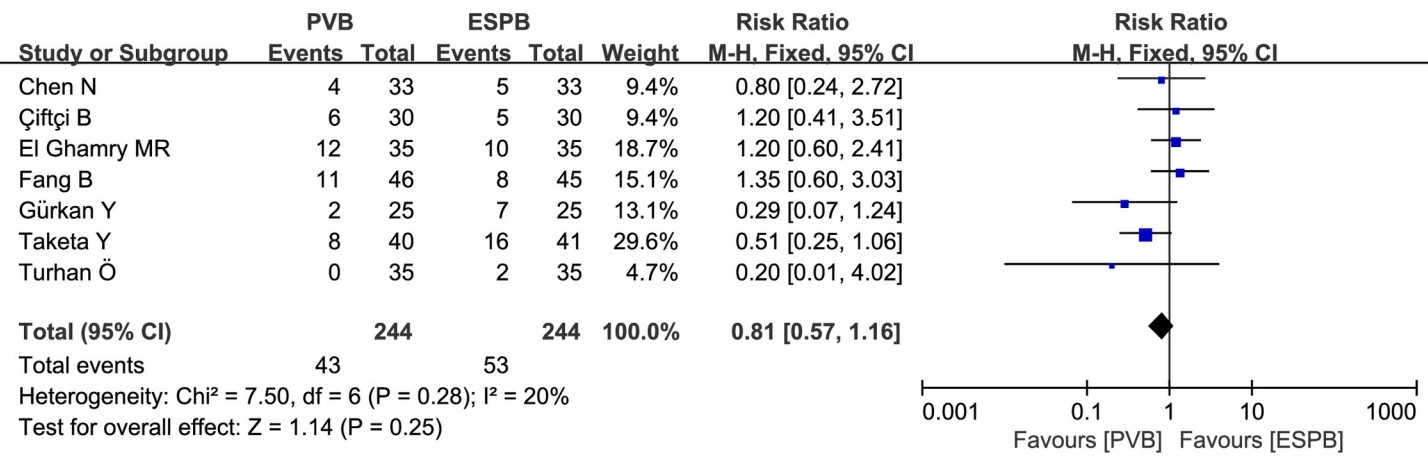

**Fig 9. Forest plot of PONV.**

## Publication bias

No substantial asymmetry was detected throughout visual examination of the funnel plot (**S1 Fig**). Also, the Egger's test showed that the P-value for each outcome was greater than 0.1, indicating no publication bias existed in this meta-analysis (**S1 Table**).

## Discussion

This meta-analysis incorporated ten RCTs encompassing 726 patients, and it aimed to directly assess postoperative analgesic effects of PVB and ESPB for thoracic and breast surgery. In recent years, due to the risks and limitations of thoracic epidural anesthesia (TEA), such as limited inclusion of patients using anticoagulants and the high rates of respiratory depression, hypotension and postoperative urinary retention associated with TEA [28,29], it has been gradually replaced by other techniques, such as ultrasound-guided PVB and ESPB. PVB has been used in clinical practice for more than 100 years, and its unilateral block features retain the sympathetic nerve function on the contralateral side, thereby greatly reducing the risk of hypotension, respiratory depression, and urinary retention [30]; moreover, a large number of studies have shown that the analgesic effect of PVB is comparable to TEA [8,31]. Therefore, an increasing number of hospitals recommend using PVB for postoperative analgesia for patients undergoing thoracic and breast surgery. However, pneumothorax, block site infection, long operation time, and high difficulty make some inexperienced anesthesiologists choose to abandon PVB or use other techniques, such as ESPB.

ESPB is a novel nerve-blocking technique first proposed by Forero et al. in 2016 [32]. It is generally implemented through deposition of drugs into the fascial plane beneath the erector spinae muscle at the tip of the transverse process of the vertebra, thereby reducing pneumothorax and significant neurovascular damage. Many studies have also reported that ESPB plays an effective role in postoperative analgesia for thoracic and breast surgery, and it is proven to be easily implemented with a high successful block rate [33,34]. A report of 242 patients also showed that ESPB had almost no operation failure or adverse complications [35]. Eventually, considering the convenience, effectiveness, and security, we conducted this meta-analysis to determine whether ESPB can become a new technique to replace PVB.

The results of this meta-analysis showed that compared with ESPB, PVB significantly reduced the postoperative pain scores at rest at 0–1, 4–6, and 24 hours for patients undergoing thoracic and breast surgery. No significant difference was found in opioid consumption at 24

hours post-operation and incidence of PONV between the two groups. Since breast surgery did not report the outcomes of the postoperative pain scores at movement and the incidence of additional analgesia in 24 hours post-operation, this meta-analysis showed PVB significantly reduced the postoperative pain scores at movement at 4–6, 8–12, and 24 hours for patients undergoing thoracic surgery as well as incidence of additional analgesia in 24 hours post-operation. Furthermore, subgroup analysis found that the postoperative scores for resting pain at 0–1, 4–6, and 24 hours and opioid consumption at 24 hours after surgery were significantly lower in the PVB group for thoracic surgery. While no significant differences were found between the two groups for breast surgery. Consequently, we can conclude that the analgesic effect of PVB is stronger than that of ESPB after thoracic surgery in terms of reducing postoperative pain scores, usage of opioid 24 hours after surgery, and incidence of additional analgesia. While the effects were similar between the two groups after breast surgery. The results were not completely consistent with the conclusion of a previous meta-analysis [36], which showed that the PVB analgesic efficacy was like that of ESPB in pain scores and opioid consumption at 24 hours after surgery. This discrepancy may be explained by the following two reasons: the included trials in this study outnumber those of the previous meta-analysis; this study analyzed thoracic surgery and breast surgery separately while the previous study mixed the two surgeries for analysis. As for PVB and ESPB showing different analgesic effects in thoracic and breast surgery, we speculate that it may be related to the insufficient analgesia provided by ESPB for thoracic surgery.

In this meta-analysis, we also analyzed the block procedure time, and the result clearly showed that the time required for ESPB was significantly shorter than that required for PVB. This is the advantage of the new technology ESPB, which reflects its simplicity to a certain extent. Thus, when choosing postoperative analgesia, convenience also plays a very important role under the premise of satisfying effectiveness and safety. So, according to the results of this meta-analysis, we suggest that in thoracic surgery, it is best to have an experienced anesthesiologist perform a PVB to achieve postoperative analgesia, which can significantly reduce postoperative pain scores, postoperative usage of opioid, and the rate of additional analgesia. However, in the selection of analgesia after breast surgery, ESPB can be considered first, which can not only achieve sufficient analgesia, but also can be completed efficiently.

Regarding heterogeneities, we initially assumed that different surgical sites were an important source. Although we conducted a corresponding subgroup analysis that reduced the heterogeneity to a certain extent, especially in the breast surgery subgroup, there was still significant heterogeneity in some of the thoracic surgery subgroups. Several of the following reasons may be the sources of the heterogeneity existing in this study: differences in experience and proficiency of the doctors who performed the block, large operation time span, differences in types and concentrations of local anesthetics, errors in data conversion, and differences in assessment of pain scores.

This meta-analysis also has several limitations. Firstly, there are only ten RCTs included in this meta-analysis, which may weaken the conclusion. Secondly, we did not analyze other adverse reactions other than PONV because of the scarcity of reported data. Lastly, only English literature was included in this study, which may contain a selection bias. Therefore, large-scale, multicenter, prospective, double-blinded RCTs are recommended to explicitly discern the effectiveness of PVB and ESPB.

## Conclusion

The postoperative analgesic effects of PVB versus ESPB are distinguished by the surgical site. For thoracic surgery, the postoperative analgesic effect of PVB is better than that of ESPB; for

breast surgery, the postoperative analgesic effects of PVB and ESPB are similar. At the same time, there is no significant statistical difference between the two postoperative analgesic techniques in PONV, but the block procedure time required for ESPB is significantly shorter than that for PVB. Therefore, when choosing postoperative analgesia techniques, the results of this meta-analysis recommend PVB for thoracic surgery and ESPB for breast surgery.

## Supporting information

**S1 Checklist. PRISMA 2009 checklist.**
(DOC)

**S1 Fig. Publication bias assessed by funnel plot.** Panel A presents the funnel plot for pain scores at rest at 0–1, 4–6, 8–12 and 24 hours, panel B for pain scores at movement at 0–1, 4–6, 8–12 and 24 hours, panel C for opioid consumption at 24 hours after surgery, panel D for incidence of additional analgesia in 24 hours post-operation, panel E for time required for completing block procedure and panel F for incidence of PONV.
(TIF)

**S1 Table. Publication bias for each outcome.**
(DOC)

**S1 File. PROSPERO protocol.**
(PDF)

**S2 File. The full search strategy of PubMed.**
(DOC)

## Acknowledgments

We thank TopEdit (www.topeditsci.com) for its linguistic assistance during the preparation of this manuscript.

## Author Contributions

**Data curation:** Chang Xiong, Chengpeng Han, Dong Zhao, Duojia Xu, Zhijian Lan.

**Formal analysis:** Chang Xiong, Chengpeng Han, Dong Zhao, Duojia Xu, Zhijian Lan.

**Funding acquisition:** Chang Xiong.

**Investigation:** Chang Xiong, Zhijian Lan.

**Methodology:** Chang Xiong, Chengpeng Han, Wenyong Peng.

**Resources:** Duojia Xu.

**Software:** Chang Xiong, Chengpeng Han, Dong Zhao, Wenyong Peng.

**Supervision:** Chengpeng Han, Zhijian Lan.

**Validation:** Wenyong Peng.

**Visualization:** Zhijian Lan.

**Writing – original draft:** Chang Xiong, Chengpeng Han.

**Writing – review & editing:** Chang Xiong, Zhijian Lan.

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
