## [Decision Letter · Decision Letter 0]

14 Jul 2021

PONE-D-21-14032

Postoperative analgesic effects of paravertebral block versus erector spinae plane block for thoracic and breast surgery: a meta-analysis

PLOS ONE

Dear Dr. Zhijian Lan 

Thank you for submitting your manuscript to PLOS ONE. After careful consideration, we feel that it has merit but does not fully meet PLOS ONE’s publication criteria as it currently stands. Therefore, we invite you to submit a revised version of the manuscript that addresses the points raised during the review process.

I would appreciate to pay a careful attention in your reply to the reviewers' comments and reply point by point to the comments in your response.  

We look forward to receiving your revised manuscript.

Kind regards,

Ehab Farag, MD FRCA FASA

Academic Editor

PLOS ONE

Journal Requirements:

2. We note that this manuscript is a systematic review or meta-analysis; our author guidelines therefore require that you use PRISMA guidance to help improve reporting quality of this type of study. Please upload copies of the completed PRISMA checklist as Supporting Information with a file name “PRISMA checklist

Reviewers' comments:

Reviewer's Responses to Questions

**Comments to the Author**

1. Is the manuscript technically sound, and do the data support the conclusions?

Reviewer #1: Partly

Reviewer #2: Yes

2. Has the statistical analysis been performed appropriately and rigorously? 

Reviewer #1: I Don't Know

Reviewer #2: Yes

3. Have the authors made all data underlying the findings in their manuscript fully available?

Reviewer #1: Yes

Reviewer #2: Yes

4. Is the manuscript presented in an intelligible fashion and written in standard English?

Reviewer #1: Yes

Reviewer #2: Yes

5. Review Comments to the Author

Reviewer #1: General comments: The overall benefit of PVB vs ESPB is found to be statistically significant but are the observed reduction in pain scores and especially opioid consumption clinically significant?

I would describe the nerve block technique in a different word than using puncture, such as time to completion of block or procedure. Puncture implies injuring the nerve or organ causing a wound. Puncture is used throughout the manuscript.

Introduction:

Line 84 – “high incidence of … respiratory depression…” the article (7) cited here actually concludes that TEA has a low incidence of major complications. Additionally, TEA can have a beneficial effect by reducing pulmonary dysfunction after thoracic or major abdominal surgery.

Line 97 – citation 11 is an editorial that does not describe data suggesting reduction in analgesic use or pain scores with ESPB.

In the introduction, there is no description of what the ESP block actually is or why does it have significantly reduced block difficulty? Should briefly describe technique and why it is different from PVB or cite from an article that describes ESP technique.

Discussion:

Line 316: “…high rates of respiratory depression,…” The article cited (28) states that TEA reduces pulmonary dysfunction following thoracic or major abdominal surgery and can be safely used in patient with compromised respiration. Reference 29 states “Good analgesia from an epidural can result in early extubation, better ventilatory mechanics and gas exchange and reduced rates of lung collapse, pneumonia and pain”.

Line 322: urine storage – would change to urinary retention

Line 326, 336, 369: would change the word puncture

Line 361: Discusses results of this study compared to previous meta-analysis (36) analyzing ESP vs PVB and how there is a discrepancy between results. Is there a reason that this study did not include the trials from cited study? Mona 2020, Yavuz 2020, Nan 2020, Yasuko 2019, Matthew 2020, Hong 2020 which all compare ESP vs PVB for breast or thoracic surgery.

Figures:

There are no descriptions for any of the figures listed

Reviewer #2: Xiong’s team performed a nice meta-analysis coming the analgesic effect of PVB and ESPB. The manuscript is nicely structured and easy to follow. I just have a few concerns that I hope the authors could address.

Major comments:

1. The aggregation of postoperative pain data. The 10 RCTs included in this meta-analysis measure pain scores at different time points or time intervals, and not all reported pain score at rest/at movement. In the write-up of the result section, the authors should clearly state how many studies reported relevant data for each time point assessed and how the data are aggregated. (e.g, some studies reported pains core and both 4 hrs and 6 hrs postop, how do you calculate pain score 4-6 hrs?) It might be clearer, if outcomes in Table 1 can be a bit more detailed.

2. In the subgroup analysis for pain score comparisons, I-squared still indicated heterogeneity for some of the comparisons – is sensitivity analysis performed to see if any study caused heterogeneity?

Minor comments:

1. Ln 223-225, line 230-232. When reporting difference at multiple time points, place the results immediately after each time point to make it easier to follow.

2. Result section: Why choosing to report pain scores at 0-1h, 4-6h, 8-12h and 24h? Several studies reported pain score at 2 hrs postop, are these data utilized?

6. PLOS authors have the option to publish the peer review history of their article (what does this mean?). If published, this will include your full peer review and any attached files.

Reviewer #1: No

Reviewer #2: No

---

## [Author Response · Author response to Decision Letter 0]

26 Jul 2021

Dear Editors and Reviewers:

I am very grateful to you for giving me an opportunity of minor revision, and I also cherish this chance. In view of your questions, my team and I have made a lot of efforts to improve our articles to meet your requirements during the past few days. The following is the reply to your suggestions.

Response to Journal Requirements:

1.We have tried our best to revise the manuscript to meet PLOS ONE's style requirements, including those for file naming. However, we hope the editor could review it again because we can't guarantee that the revised manuscript will fully meet the requirements of the journal.

2.We have uploaded the completed PRISMA checklist as Supporting Information with a file name “PRISMA checklist”.

3.According to the problems pointed out by reviewer 1, we have replaced references (7), (11), (28), (29).

Response to Reviewers:

Reviewer #1: 

General comments: The overall benefit of PVB vs ESPB is found to be statistically significant but are the observed reduction in pain scores and especially opioid consumption clinically significant?

Answer: Generally, pain scores are divided into three levels: 1-3 as mild, 4-6 as moderate, and 7-10 as severe. Most patients can tolerate mild-level pain, which generally does not have a significant impact on the body. Moderate- and severe-level pain often need to be controlled. In our study, additional analgesia was used when the pain scores were ≥4. In fact, current studies rarely report severe pain, thanks to the two effective analgesic techniques PVB and ESPB, as well as various intravenous analgesics. These control the postoperative pain scores of some patients at a mild level, which introduces difficulty in showing the difference between PVB and ESPB in these patients. We believe that the decrease in the pain scores of PVB compared with ESPB in thoracic surgery is closely related to the reduction of the scores of patients with moderate- and sever-level pain. In other words, fewer patients who use PVB analgesia experience moderate- or higher level pain than those who use ESPB, which is confirmed by the incidence of additional analgesia. Similarly, the amount of opioid consumption in the PVB group within 24 hours after surgery decreased by about 2.74 mg, compared with the ESPB group; if corresponds to patients with moderate- and severe-level pain scores, this value may increase. More importantly, we think your question is very constructive, and we are considering designing a larger sample size, more detailed and more targeted trial to verify the above conjecture. 

I would describe the nerve block technique in a different word than using puncture, such as time to completion of block or procedure. Puncture implies injuring the nerve or organ causing a wound. Puncture is used throughout the manuscript.

Answer: After reading the relevant literature, we have accepted your proposal and amended “puncture” to “block procedure” or “block”. And these changes were marked in red throughout the manuscript. 

Introduction:

Line 84 – “high incidence of … respiratory depression…” the article (7) cited here actually concludes that TEA has a low incidence of major complications. Additionally, TEA can have a beneficial effect by reducing pulmonary dysfunction after thoracic or major abdominal surgery.

Answer：After careful reading, we replaced Reference (7). 

Line 97 – citation 11 is an editorial that does not describe data suggesting reduction in analgesic use or pain scores with ESPB.

Answer：After careful reading, we also replaced Reference (11). 

In the introduction, there is no description of what the ESP block actually is or why does it have significantly reduced block difficulty? Should briefly describe technique and why it is different from PVB or cite from an article that describes ESP technique.

Answer: Thank you for this suggestion. Accordantly, we have added a brief description of ESPB in the introduction section at lines 94-99.

Discussion:

Line 316: “…high rates of respiratory depression,…” The article cited (28) states that TEA reduces pulmonary dysfunction following thoracic or major abdominal surgery and can be safely used in patient with compromised respiration. Reference 29 states “Good analgesia from an epidural can result in early extubation, better ventilatory mechanics and gas exchange and reduced rates of lung collapse, pneumonia and pain”.

Answer: We also replaced references 28 and 29. Thank you very much for pointing out the errors in the references. 

Line 322: urine storage – would change to urinary retention

Answer: We have changed “urine storage” to “urinary retention” in the manuscript.

Line 326, 336, 369: would change the word puncture

Answer: We have changed “puncture” to “block procedure” or “block” in the manuscript.

Line 361: Discusses results of this study compared to previous meta-analysis (36) analyzing ESP vs PVB and how there is a discrepancy between results. Is there a reason that this study did not include the trials from cited study? Mona 2020, Yavuz 2020, Nan 2020, Yasuko 2019, Matthew 2020, Hong 2020 which all compare ESP vs PVB for breast or thoracic surgery.

Answer: The previous meta-analysis showed that the PVB analgesic efficacy was like that of ESPB in pain scores and opioid consumption at 24 hours after surgery. But this study found that the postoperative scores and opioid consumption at 24 hours after surgery were significantly lower in the PVB group than that in ESPB group for thoracic surgery. We have discussed it in the discussion section at lines 362-373.

The studies of Mona 2020, Yavuz 2020, Nan 2020, Yasuko 2019, Matthew 2020, Hong 2020 included in the previous meta-analysis (Ref 36) are actually included in this study, but the naming method is different. For example, "Mona 2020", the full name of the first author of this study is "Mona Raafat El Ghamry", and the name given in PUBMED is "El Ghamry MR", therefore, we cite "El Ghamry MR 2020" in this study. Their relationship is as follows：

Mona 2020 = El Ghamry MR 2019 (Ref 20) (this study was indeed published in 2019); 

Yavuz 2020 = Gürkan Y 2020 (Ref 22); 

Nan 2020 = Chen N 2020 (Ref 18);

 Yasuko 2019 = Taketa Y 2019 (Ref 25); 

Matthew 2020 = Swisher MW 2020 (Ref 24); 

Hong 2020 = Zhao H 2020 (Ref 27).

Figures:

There are no descriptions for any of the figures listed.

Answer: According to the journal requirements, each figure caption was showed directly after the paragraph in which they are first cited.

Reviewer #2: 

Major comments:

1. The aggregation of postoperative pain data. The 10 RCTs included in this meta-analysis measure pain scores at different time points or time intervals, and not all reported pain score at rest/at movement. In the write-up of the result section, the authors should clearly state how many studies reported relevant data for each time point assessed and how the data are aggregated. (e.g, some studies reported pains core and both 4 hrs and 6 hrs postop, how do you calculate pain score 4-6 hrs?) It might be clearer, if outcomes in Table 1 can be a bit more detailed.

Answer: In accordance with the reviewer’s comment, we have added relevant information in the results section at lines 203-207. Moreover, we have also added “Time to assess pain scores” in Table 1.

2. In the subgroup analysis for pain score comparisons, I-squared still indicated heterogeneity for some of the comparisons – is sensitivity analysis performed to see if any study caused heterogeneity?

Answer: Based on the subgroup analysis, it can be seen that the heterogeneity was significantly decreased in the breast surgery subgroup, while remained in the thoracic surgery subgroup. By conducting a sensitivity analysis omitting one study by turns on the subgroup of thoracic surgery, we found that the significant heterogeneity was still present among the trials. As mentioned in the write-up of the discussion section, there may be many factors that produce heterogeneity, such as differences in experience and proficiency of the doctors who performed the block, large operation time span, differences in types and concentrations of local anesthetics, errors in data conversion, and differences in assessment of pain scores.

Minor comments:

1. Ln 223-225, line 230-232. When reporting difference at multiple time points, place the results immediately after each time point to make it easier to follow.

Answer: We accepted your comments and revised all of them in the manuscript. (At line 49-57, 233-235, 240-243, 249-252, 254-258).

2. Result section: Why choosing to report pain scores at 0-1h, 4-6h, 8-12h and 24h? Several studies reported pain score at 2 hrs postop, are these data utilized?

Answer: At the beginning, we planned to report pain scores at 0-2 h, 4-6 h, 8-10 h, and 24 h after surgery. We believe that the pain scores within such time intervals will not have much difference, which helps to continuously observe the changes in postoperative pain. Only 4 studies reported the pain scores at 2 h postoperatively. Therefore, we did not use the data, and adjusted the time interval from 0-2 h to 0-1 h after surgery. Meanwhile, we adjusted the time interval from 8-10 h to 8-12 h to include more research data, for which the pain scores of 8-10 h and 10-12 h are relatively close and can be analyzed together.

Finally, I would like to extend my sincere wishes to the editor and the reviewer for a healthy life and a smooth career.

---

## [Editor Report · Decision Letter 1]

11 Aug 2021

Postoperative analgesic effects of paravertebral block versus erector spinae plane block for thoracic and breast surgery: a meta-analysis

PONE-D-21-14032R1

Dear Dr. Zhijian Lan 

We’re pleased to inform you that your manuscript has been judged scientifically suitable for publication and will be formally accepted for publication once it meets all outstanding technical requirements.

Kind regards,

Ehab Farag, MD FRCA FASA

Academic Editor

PLOS ONE
---

## [Editor Report · Acceptance letter]

13 Aug 2021

PONE-D-21-14032R1 

Postoperative analgesic effects of paravertebral block versus erector spinae plane block for thoracic and breast surgery: a meta-analysis 

Dear Dr. Lan:

I'm pleased to inform you that your manuscript has been deemed suitable for publication in PLOS ONE. Congratulations! Your manuscript is now with our production department. 

Kind regards, 

on behalf of

Dr. Ehab Farag 

Academic Editor

PLOS ONE